# Coating of Filter Materials with CeO_2_ Nanoparticles Using a Combination of Aerodynamic Spraying and Suction

**DOI:** 10.3390/nano13243168

**Published:** 2023-12-18

**Authors:** Anna V. Abramova, Daniil A. Kozlov, Varvara O. Veselova, Taisiya O. Kozlova, Olga S. Ivanova, Egor S. Mikhalev, Yuri I. Voytov, Alexandr E. Baranchikov, Vladimir K. Ivanov, Giancarlo Cravotto

**Affiliations:** 1Kurnakov Institute of General and Inorganic Chemistry, Russian Academy of Sciences, Leninsky Prospekt 31, 119991 Moscow, Russia; kozlov@inorg.chem.msu.ru (D.A.K.); ibvarvara@yandex.ru (V.O.V.); taisia.shekunova@yandex.ru (T.O.K.); uri01@mail.ru (Y.I.V.); van@igic.ras.ru (V.K.I.); 2Frumkin Institute of Physical Chemistry and Electrochemistry, Russian Academy of Sciences, Leninsky Prospekt 31, 119991 Moscow, Russia; runetta05@mail.ru; 3Limited Liability Company “Angstrem”, Bolshaya Polyanka, 51A/9, 119180 Moscow, Russia; mikhalevec@gmail.com; 4Dipartimento di Scienza e Tecnologia del Farmaco, University of Turin, Via P. Giuria 9, 10125 Turin, Italy

**Keywords:** nanostructured coatings, cerium oxide nanoparticles, antibacterial activity, aerodynamic acoustic emitter

## Abstract

Textiles and nonwovens (including those used in ventilation systems as filters) are currently one of the main sources of patient cross-infection. Healthcare-associated infections (HAIs) affect 5–10% of patients and stand as the tenth leading cause of death. Therefore, the development of new methods for creating functional nanostructured coatings with antibacterial and antiviral properties on the surfaces of textiles and nonwoven materials is crucial for modern medicine. Antimicrobial filter technology must be high-speed, low-energy and safe if its commercialization and mass adoption are to be successful. Cerium oxide nanoparticles can act as active components in these coatings due to their high antibacterial activity and low toxicity. This paper focuses on the elaboration of a high-throughput and resource-saving method for the deposition of cerium oxide nanoparticles onto nonwoven fibrous material for use in air-conditioning filters. The proposed spraying technique is based on the use of an aerodynamic emitter and simultaneous suction. Cerium oxide nanoparticles have successfully been deposited onto the filter materials used in air conditioning systems; the antibacterial activity of the ceria-modified filters exceeded 4.0.

## 1. Introduction

The development of new methods for creating functional nanostructured coatings on the surfaces of materials of various physical natures is extremely relevant, with such coatings currently being used in almost all areas of industry.

Healthcare-associated infections (HAIs) affect 5–10% of patients in hospitals and are the tenth leading cause of death [1,2,3]. These infections include tuberculosis, chickenpox, measles, flu, severe acute respiratory syndromes (SARS) and methicillin-resistant staphylococcus aureus (MRSA) [4]. Some patient groups are particularly vulnerable: newborns, the elderly, patients undergoing aggressive and invasive medical manipulations and organ transplants, etc. In these groups, the incidence of healthcare-associated infections is significantly higher. This problem was especially acute during the COVID-19 pandemic. A notable proportion of HAIs are spread via airborne transmission from patient to patient, from medical personnel to patient and through ventilation systems, with relevant diseases including tuberculosis, chickenpox, measles, flu [4], SARS and methicillin-resistant *Staphylococcus aureus* (MRSA).

Patients and healthcare workers produce bioaerosols when they breathe, talk, sneeze and cough [5,6]. Medical interventions, such as intubation, are also a source of bioaerosols [7]. All of these actions create droplets that can stay in the air for a long period of time and travel through a hospital environment via convection motion [4]. Moreover, droplets can evaporate to form nuclei (<5 µm) and remain infectious [8]. Air filtration and disinfection are, therefore, critically important in medical organizations. However, the nonwoven materials used in ventilation systems are one of the main transmitters of cross-infection in patients [1,2]. Poorly maintained air conditioning systems become reservoirs of dangerous bacteria (e.g., legionella), which can be further aerosolized into the air. This leads to a high risk of infection via inhalation and a high incidence of lethality [4]. Ventilation, dehumidification and air conditioning systems are often equipped with multistage airflow filtration systems. The main load falls on the multilayer primary filter, which removes dust, soot and dust mites from the air. Antibacterial coatings may be able to protect these filters from biocontamination.

Antibiotics have played a key role in managing bacterial infections over the last century. However, the overuse and misuse of antibiotics have led to the emergence of multidrug-resistant (MDR) bacteria, biofilms, etc. While evaluating the harm caused by MDR organisms is complicated, MDR infections, for example, were named the third highest cause of death in the United States in 2010 [9]. The recent and excessive consumption of various types of antibiotics during the COVID-19 pandemic [10] and their considerable leakage into the environment [11] have resulted in further unwanted antibiotic exposure around the world and exacerbated MDR evolution.

Conventional antibiotics have lost the majority of their worth, and new therapeutic platforms are being developed. Certain types of nanomaterials, such as fullerene derivatives [12], gold nanoparticles [13,14,15,16], ferromagnetic nanoparticles [17], rare earth nanoparticles [18,19,20,21,22,23,24] and others, have been reported to possess antibacterial properties and have been observed to be highly effective against MDR strains.

One of the key demands for the practical use of antibacterial nanoparticles is their safety for humans and the environment. Cerium oxide (CeO_2_) nanoparticles have excellent antibacterial activity against both Gram-negative and Gram-positive bacteria [25,26,27,28,29,30,31,32], exhibit antiviral properties [33,34,35,36] and possess low cytotoxicity to mammalian cells [37,38]. The particle size of CeO_2_ greatly affects its antibacterial [39] and antiviral [40] properties [41,42,43], which allows the desired effect to be fine-tuned. It is worth noting that cerium (Ce) is one of the most abundant rare earth elements, making the nanomaterials produced using CeO_2_ highly affordable.

The development of new effective methods for the coating of various materials is extremely relevant. There are various methods for applying antimicrobial particles to nonwovens [44], including high-frequency low-pressure induction (HFI) plasma [45], which deposits silver nanoparticles in an environmentally friendly way, but suffers from quite low productivity.

The successful commercialization and mass adoption of antimicrobial filter technology require it to be high-speed, low-energy and safe. In this regard, the sonochemical method [46,47] is a promising alternative for depositing nanoparticles on fibers from suspensions. The physical and chemical phenomena caused by cavitation in liquids [48,49,50] allow in situ syntheses to be carried out and metal or metal oxide nanoparticles to be deposited on textiles or nonwoven materials. Cavitation-induced shock waves and microjets in liquids cause nanoparticles to collide with textile surfaces at high speeds, resulting in strong nanoparticle adhesion to the fibers and the formation of stable coatings [45]. However, our previous findings show that the process is quite inefficient for nonwovens. 

The dip-rolling process from liquid has been widely used in various high-throughput industrial plants for the deposition of nanoparticles. Although this method of nanoparticle deposition is simple and fast, it has several drawbacks when used in different industrial plants. One of the main disadvantages is the lack of precise control over the thickness and uniformity of the nanoparticle coating. This can lead to variations in coating thickness across the filters, which may not meet the stringent requirements for antibacterial applications. Multiple dip and roll cycles are often required to achieve the desired coating thickness, which can increase production time and cost, making it less suitable for high-throughput manufacturing. In addition, the dip-rolling process can lead to inefficiency and waste of nanoparticle materials. 

In addition, the need for liquid processing limits the performance of the method and requires the use of additional reagents and the additional drying of the material, thus increasing the cost of the final product. Thus, many industrial sites choose the spraying technique for their coating processes [51].

This current manuscript presents a spraying method to produce nanoparticle coatings on nonwovens that does not involve immersing the material in a liquid. The disadvantages of the classical spraying method [52] are its lack of applicability on multilayer materials (for example, for filters) due to the impossibility of ensuring the penetration of nanoparticles into the deeper layers of the material and the unevenness of the resulting coating where a nanosol is used, which is due to the fact that the sprayed nanoparticle suspension forms droplets of various sizes during spraying.

In this study, the sputtering of cerium oxide nanoparticles was achieved using aerodynamic acoustic emitters with simultaneous suction from the reverse side of the material. This experimental design ensures the penetration of the droplets, and hence of the nanoparticles, into the bulk of the material due to the pressure difference. At the same time, the method makes it possible to control the size of the droplets.

## 2. Materials and Methods

### 2.1. Synthesis

The hydrothermal synthesis of ultrafine CeO_2_ nanoparticles from ceric ammonium nitrate (NH_4_)_2_Ce(NO_3_)_6_ is described in detail by Shcherbakov et al. [53]. In brief, 2.33 g of (NH_4_)_2_Ce(NO_3_)_6_ was dissolved in 23 mL of distilled water and heated at 95 °C for 24 h. The resulting light-yellow precipitate was separated by centrifugation, washed three times with isopropanol and redispersed in 25 mL of deionized water. To remove residual isopropanol, the solution was boiled for 1 h. The resulting sol was labeled “CeO_2_”.

A citrate ion-stabilized CeO_2_ sol was synthesized from cerium (III) chloride and citric acid. In brief, 2.33 g of CeCl_3_·7H_2_O and 4 g of citric acid were dissolved in 200 mL of a water–isopropanol solution (V(^i^PrOH):V(H_2_O) = 20:1) and kept under vigorous stirring for 2 h. The resulting white precipitate was separated via filtration, washed with isopropanol and dried at 50 °C overnight. The dried powder was dispersed in 100 mL of distilled water, and 10 mL of concentrated aqueous ammonia (28 wt.%) was then added. The obtained solution was stirred for 2 h at room temperature and heated at 95 °C for 6 h until a dark-brown sol formed. During heating, the pH value of the solution was kept higher than 12 via the addition of ammonia solution. At the final stage of the synthesis, the sol was boiled to get rid of the excess ammonia and to lower the pH of the sol to ~8. The resulting sol was labeled “CeO_2_-Cit”. The exact concentration of cerium oxide was determined gravimetrically.

The sol was dispersed onto the surface of the Topperr (Russia) FC 1 textile air conditioner filter 350 × 700 mm^2^ using an experimental setup, the schematic diagram of which is shown in Figure 1. The experimental equipment consisted of a compressor, a vacuum pump, an aerodynamic acoustic emitter, a cylindrical transparent vessel, a vacuum gauge, a vessel for an aqueous sol, a separating plate and the filter to be processed. The separating plate divided the cylindrical vessel into two zones: a vacuum pump created a low-pressure area on the back side of the filter (in the upper zone of the container, see Figure 1), and the sol was sprayed from the bottom (see the position of the hydrodynamic acoustic emitter in Figure 1). The aerodynamic acoustic emitter was supplied with compressed air at a pressure of 4 atm.

The working principle of the proposed method is the generation of microdroplets of sol using an aerodynamic emitter. This technique allows the distribution of the droplet size to be controlled. The mist forms below the filter and is sucked through the filter by the vacuum pump. By using the aerodynamic emitter, we can ensure that the size of the droplets is small enough to penetrate between the fibers and reach the back of the filter. With this technique, we achieve coverage of the fibers on both the front and back sides of the filter. This ensures the antibacterial properties of the whole filter and contributes to preventing the development of zones where bacteria could form colonies due to the lack of coating.

In order to deposit cerium oxide onto the filter surface, the synthesized sols were diluted with distilled water and isopropanol to achieve CeO_2_ concentrations of 2.5 g/L and an isopropanol volume fraction of 40%.

### 2.2. Methods

Powder X-ray diffraction (PXRD) patterns were acquired using the powder diffractometer D8 Advanced (Bruker, Berlin, Germany) in the reflection geometry (Bragg–Brentano) with CuKα_1,2_ radiation. XRD patterns were collected in the 20–80° 2θ range with a 0.02° step. The identification of the diffraction peaks was carried out using the ICDD database (PDF2, release 2020). PXRD pattern refinements were performed using the Rietveld method in Maud software [54]. Scanning electron microscopy (SEM) images were obtained using NVision 40 (Zeiss, Oberkochen, Germany) and Amber GMH (Tescan, Brno, Czech Republic) microscopes operated at an accelerating voltage of 1–2 kV, using secondary and backscattered electron detectors. Energy-dispersive X-ray spectroscopy (EDS) was performed using an Ultim Max (Oxford Instruments, Abingdon, UK) detector at an accelerating voltage of 20 kV. Transmission electron microscopy (TEM) images were acquired using a charge-coupled device (CCD) camera, Ultra Scan 4000 (Gatan, Pleasanton, CA, USA), installed in a transmission electron microscope, Libra 200MC (Zeiss, Oberkochen, Germany), operated at 200 kV. Electron energy loss (EEL) spectra were collected in the conventional TEM mode. The half-width of the peak of zero electron energy loss was 0.2 eV. The EEL spectra were processed using Digital Micrograph software.

The ceria sol absorption spectra were measured in the wavelength range of 200–800 nm with a 0.1 nm resolution using an SF-2000 (OKB Spectrum, Saint-Petersburg, Russia) spectrometer. The optical absorption spectra of the filters after CeO_2_ deposition were collected on a QE65000 (Ocean Optics, Dunedin, FL, USA) spectrometer in the 200–1200 nm wavelength range using an ISP 50 8 R (Ocean Optics, Dunedin, FL, USA) integrating sphere. A combination of deuterium and halogen lamps was used as a light source in the DH-2000 (Ocean Optics, Dunedin, FL, USA) device.

A suspension of Escherichia coli K12 with an optical density of OD = 0.8 was used to assess the antibacterial properties of the coatings. The fabric samples were placed in the culture medium LB + 1.5% agar. Subsequently, they were coated with 100 μL of bacterial suspension. After 12 h of incubation at 37 °C, the fabric samples were removed from the medium, and the residual antibacterial effect was tested on petri dishes. The test was carried out with four repetitions for every type of sample. The quantitative characteristics of antibacterial activity were analyzed using the standard method ISO 20743:2012. The antibacterial performance of the samples was evaluated using the formula A = F − G, where F is the bacterial growth rate of the control samples (log10 CFU/mL after incubation–log10 CFU/mL prior to incubation), CFU is colony-forming units and G is the bacterial growth rate of the test samples.

## 3. Results

Optical absorption spectroscopy showed that both the CeO_2_ and CeO_2_-Cit sols possess an absorption edge in the range of 300–400 nm, and no absorption bands at longer wavelengths were detected (Figure 2A). A clear shift in the absorption edge of the CeO_2_-Cit sol towards the long-wavelength region is most likely caused by an admixture of trivalent cerium compounds, both on the surface and in the structure of the cerium dioxide nanoparticles. 

According to the results of powder X-ray diffraction analysis, both sols contain a cerium oxide phase (ICDD PDF2 card 34-0394), with other crystalline phases being absent (Figure 2B). A full profile analysis of the diffraction patterns was performed using the Rietveld method, which showed that the region of coherent scattering (crystallite size) of cerium oxide nanoparticles is 3.0 nm for the CeO_2_-Cit sol and 3.8 nm for the CeO_2_ sol.

A detailed study of cerium oxide nanoparticles was carried out via transmission electron microscopy, and the high-resolution TEM images are shown in Figure 3A,B. In order to determine the average size, a statistical analysis of TEM images was performed, and the size distribution diagrams were obtained (Figure 3C,D). The particle sizes of the sols do not differ statistically. The cerium oxide sol that was obtained without the use of a stabilizer (CeO_2_) displayed a particle size of 2.8 ± 0.6 nm, while the sol synthesized in the presence of citric acid (CeO_2_-Cit) had a particle size of 2.5 ± 0.5 nm. The obtained values are close to the sizes of the coherent scattering regions measured using PXRD. However, since the CeO_2_-Cit sol was synthesized using the Ce^3+^ compound (CeCl_3_), residual trivalent cerium may be present in the sol, and the electron energy loss spectroscopy (EELS) method was used to determine the possible presence of Ce^3+^. The EEL spectra (Figure 3E) are similar in the region of the Ce-M4 and Ce-M5 edges of both nanoparticle assemblies. However, a significant shift in the absorption edges of Ce-M4 and Ce-M5 to lower energies indicates a larger Ce^3+^ fraction in the CeO_2_-Cit nanoparticles. At the same time, it is impossible to distinguish Ce^3+^ ions that are adsorbed onto the ceria surface from those embedded in the ceria crystal lattice. In summary, it can be argued that the differences in the CeO_2_ and CeO_2_-Cit sols are mainly due to the presence of trivalent cerium in the CeO_2_-Cit sol, while the sizes of the nanoparticles in the sols are nearly identical.

The diluted sols were used for the preparation of cerium-oxide-nanoparticle-based antibacterial coatings. The deposition of the nanoparticles onto the surface of air conditioning filters was performed using an aerodynamic emitter and a compressor to create a pressure difference between the sides of the filter. The filter can be placed at various distances from the aerodynamic emitter, and so preliminary experiments were carried out to determine the optimal distance between the emitter and the textile air conditioner filter using a solution of PRO COLOR (Russia) red dye. Figure 4 shows the photographs of the obtained samples.

At a distance of 40 cm from the emitter (Figure 4B), the filter enters the zone where the drops formed by the emitter become significantly smaller. However, at a distance of 30 cm, the flow distribution becomes nonuniform. At a distance of 50 cm (Figure 4C), the relative uniformity of the flow covering the treated area is restored, and the distances of 20 cm and 50 cm were therefore chosen for ceria nanoparticle deposition.

Next, the filter treatment modes were tested using the CeO_2_ sol by varying the deposition duration and emitter-to-filter distances. The optical absorption spectra of the filters after the deposition of the cerium oxide nanoparticles were collected in diffuse reflection mode (Figure 5); the peak at 360–400 nm in the spectra corresponds to the absorption of cerium oxide. 

A comparison of the optical absorption spectra of the filters with ceria nanoparticles deposited using different emitter-to-filter distances (20 and 50 cm) showed that the content of cerium oxide differs slightly on the treated side of the filter. However, higher contents are achieved on the back side of the filter at a distance of 50 cm after both 20 s and 60 s of treatment (Figure 5A). Depositing a cerium oxide sol on the filter at a distance of 50 cm for different times (20–150 s) shows a proportional increase in the content of cerium oxide on the treated surface (Figure 5B). At the same time, a significant increase in CeO_2_ content on the back side is observed only after treatment for 150 s, whereas treatment for 20 and 60 s shows comparable content. A similar increase in cerium oxide content was also observed when the CeO_2_-Cit sol was deposited onto the filter (Figure 5C).

An emitter-to-filter distance of 50 cm and a treatment duration of 20 s were chosen for the further production of ceria-based stable coatings on the filter surfaces. At this distance, the formation of a stable flow is achieved, and minimal differences in the cerium oxide content on the treated and back sides are ensured. Regarding the duration of treatment, multiple increases only led to slight increases in cerium oxide content on the treated and back sides, and a treatment time of 20 s was therefore chosen to reduce nanoparticle-sol consumption.

SEM images of the filter samples before and after the deposition of cerium oxide nanoparticles are shown in Figure 6. The bare fibers of the filters have a mean size of about 20–30 μm with a smooth surface (Figure 6A). The microstructure of the fibers after the deposition of CeO_2_ (Figure 6B) and CeO_2_-Cit (Figure 6C) nanoparticles at a distance of 50 cm and a duration of only 20 s was significantly altered. The ceria-based coating is observed on the fiber surfaces, as is seen in the SEM images taken in the elemental contrast mode. CeO_2_ nanoparticles completely cover the surface of the fibers, whereas individual aggregates are formed in the case of CeO2-Cit. The EDS spectra of the filters covered with CeO_2_ and CeO_2_-Cit nanoparticles (Figure 6D) confirm the presence of cerium on the surface of the fibers. The EDS spectra of CeO_2_-Cit and CeO_2_ samples contain chlorine and nitrogen, which are most likely residues from the cerium-containing precursors: cerium (III) chloride and ammonium cerium (IV) nitrate, respectively. Aluminum in the EDS spectra is detected from the SEM stage material.

The presence of nanoparticles on both sides of the filter confirms that the pressure difference on the different sides of the filter ensures the penetration of nanoparticles into the bulk of the nonwoven material. It has been found in preliminary experiments that activating the suction during the coating process leads to a significant increase in cerium oxide content on both sides of the filter, compared to cerium oxide content on filters coated by spraying without suction, even at low processing times. 

The total content of cerium oxide in the filters after treatment was assessed gravimetrically (Table 1). When CeO_2_ nanoparticles were deposited on a filter, their content doubled with increasing treatment time from 20 to 60 s, reaching 2.1 and 4.2 wt. %, respectively. Nevertheless, the deposition of CeO_2_-Cit sol at treatment times of 20 and 60 s results in a cerium oxide content of 0.7–0.8 wt. %. The lower content of ceria in the case of CeO_2_-Cit deposition may be due to the presence of the citrate anion on the nanoparticle surfaces. It has been found that an increase in the deposition duration of CeO_2_ sol leads to an increase in the content of ceria on the fibers, whereas, in the case of CeO_2_-Cit sol, the ceria content barely changes with time. Moreover, it should be noted that it is possible to achieve higher concentrations of cerium oxide on the fibers when using a distance of 50 cm between the filter and the emitter. Presumably, this is due to the fact that smaller droplets penetrate deeper into the material. Thus, the samples obtained at an emitter-to-filter distance of 50 cm and at a treatment time of 20 s were chosen for further study into biological activity.

Table 2 presents the results of the antibacterial activity studies of the obtained samples. *E. coli* bacteria were applied to the samples to check their antibacterial activity.

As can be seen from Table 2, the filters coated with ceria nanoparticles using CeO_2_ sol demonstrate quite low antibacterial activity, up to 0.45, at a ceria surface concentration of 3.4 × 10^−4^ g/cm^2^. Despite the much lower ceria concentrations in the samples coated with CeO_2_-Cit sol (1.2 × 10^−4^ g/cm^2^), their antibacterial activity is significantly higher and reaches 4.15.

The antibacterial activity of nanoparticle-based materials can be governed by various factors, including the properties of the nanoparticles and their coating approach over substrates or filter surfaces [55,56]. Taking into account the fact that the same conditions were used during nanoparticle deposition and antibacterial testing, it can be assumed that the antibacterial activity of the filters is caused by the features of the cerium oxide nanoparticles.

As shown above, the key difference between the CeO_2_ and CeO_2_-Cit samples is the higher content of Ce^3+^ in the latter, which may be the reason for its higher antibacterial activity, and the antibacterial effect of cerium compounds (+3) has recently been discussed [57]. In particular, Zhao et al. [58] have demonstrated that Ce^3+^ ions exert a dose-dependent antibacterial effect on *E. coli*, while Yin et al. [59] have produced gelatin fibers containing Ce^3+^ that also have high antibacterial activity against this type of bacteria. Sargia et al. [60] have shown that CeO_2_ nanoparticles with a high content of cerous species (Ce^3+^) on the surface have pronounced antibacterial activity against *E. coli*.

Possible approaches to increasing the CeO_2_-Cit content on the filter material are a surface modification, which would clearly influence the antibacterial activity of the nanoparticles, or a change in the application conditions. However, the results obtained showed that a significant increase in deposition time only slightly increases the concentration of sol nanoparticles, which makes the deposition process less resource-efficient. At the same time, our findings demonstrated that even lower concentrations of CeO_2_-Cit exhibit strong antibacterial activity. Therefore, further research should focus on the development of synthesis methods that ensure a higher Ce^3+^ content, since, as mentioned above, we believe that the trivalent cerium content is crucial for the antibacterial properties. 

In addition, a comprehensive study of the antibacterial performance of CeO_2_-Cit-coated filters over time is planned. To this end, filter samples will be placed in the air conditioning systems of different institutions (including hospitals) to determine the performance of the coating when exposed to large amounts of bacteria. 

## 4. Conclusions

This work proposes a method for coating nonwoven textile fine filters for air conditioners with nanoparticles. This method includes the spraying of nanoparticle colloid solutions through aerodynamic emitters in combination with a suction element at the rear of the filter. This approach makes it possible to deposit nanoparticle coatings onto nonwoven materials, e.g., filters used in air conditioning systems. 

In this work, cerium oxide nanoparticles were deposited on nonwoven materials to impart antibacterial properties. Ceria nanoparticles were synthesized using two different approaches, with either cerium (III) chloride or cerium (IV) ammonium nitrate as the precursor. An analysis of particle size, phase and chemical composition revealed that the key difference was the presence of trivalent cerium in cerium oxide nanoparticles synthesized from cerium (III) chloride and citric acid, and that it is the presence of this trivalent cerium in ceria nanoparticles that causes a significant increase in antibacterial activity. Filters coated with bare CeO_2_ nanoparticles demonstrate only slightly higher antibacterial activity than the control sample. By contrast, filters coated with citrate-stabilized cerium oxide nanoparticles with an admixture of trivalent cerium demonstrated high antibacterial properties, with the *E. coli* suppression level being over 99.9999%, giving an antibacterial activity of more than 4.15. 

We believe that the higher content of Ce^3+^ in citrate-stabilized cerium oxide could be the reason for its higher antibacterial activity. Further research should focus on the development of synthesis methods that ensure a higher content of Ce^3+^.

This method is used to produce air filters for use in healthcare facilities to prevent the spread of nosocomial infections. In this context, a comprehensive study of the antibacterial performance of CeO_2_-Cit-coated filters over time is planned. To this purpose, filter samples will be placed in the air conditioning systems of different institutions (including hospitals) to determine the performance of the coating when exposed to large amounts of bacteria.

## Figures and Tables

**Figure 1 nanomaterials-13-03168-f001:**
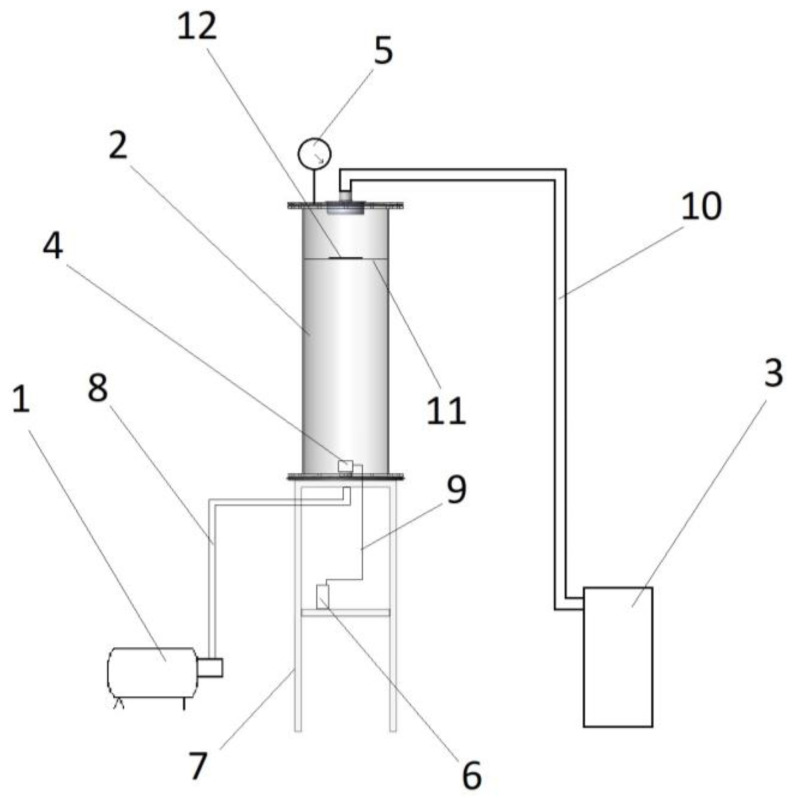
Principal scheme of the experimental setup: 1—compressor; 2—transparent cylindrical container; 3—vacuum pump; 4—aerodynamic acoustic emitter; 5—vacuum gauge; 6—vessel with sol solution; 7—base; 8—compressor pipe; 9—emitter tube; 10—vacuum pump hose; 11—separation plate; 12—filter.

**Figure 2 nanomaterials-13-03168-f002:**
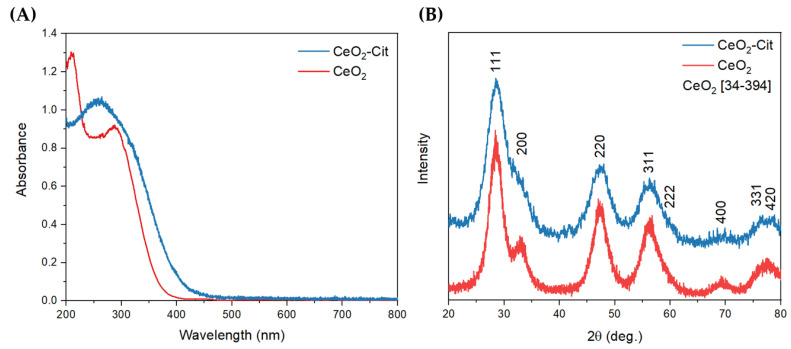
(**A**) Optical absorbance spectra of ceria sols. (**B**) PXRD patterns of ceria nanoparticles.

**Figure 3 nanomaterials-13-03168-f003:**
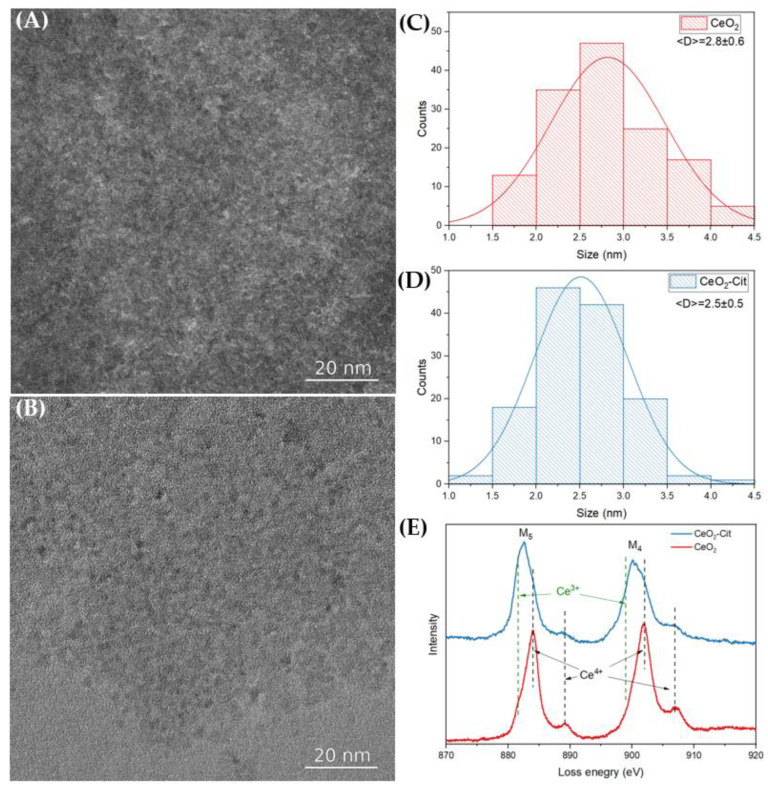
Ceria nanoparticle TEM images of (**A**) CeO_2_ and (**B**) CeO_2_-Cit sols and (**C**,**D**) corresponding nanoparticle size distributions. (**E**) Electron energy loss spectra of ceria nanoparticles: green and black dashed lines show Ce^3+^ and Ce^4+^ peak positions.

**Figure 4 nanomaterials-13-03168-f004:**
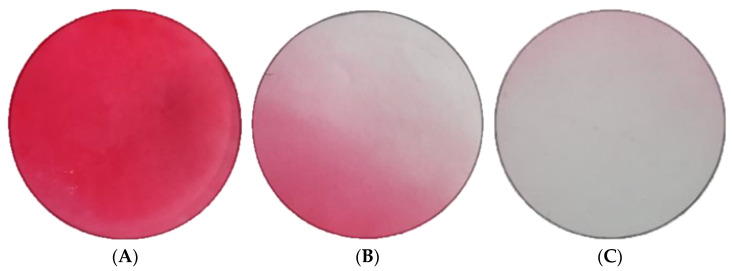
Photographs of the treated area with a diameter of 10 cm sprayed with dye at a distance of (**A**) 20 cm, (**B**) 40 cm and (**C**) 50 cm from the aerodynamic emitter.

**Figure 5 nanomaterials-13-03168-f005:**
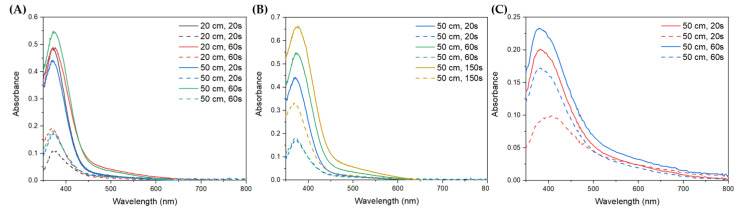
Optical absorption spectra of filters with deposited nanoparticles of CeO_2_ sol: (**A**) at different emitter-to-filter distances; (**B**) at different dispersion durations. (**C**) nanoparticles of CeO_2_-Cit sol at different dispersion durations. Solid lines correspond to spectra collected from the treated side, and the dashed lines indicate spectra obtained from the filter back side.

**Figure 6 nanomaterials-13-03168-f006:**
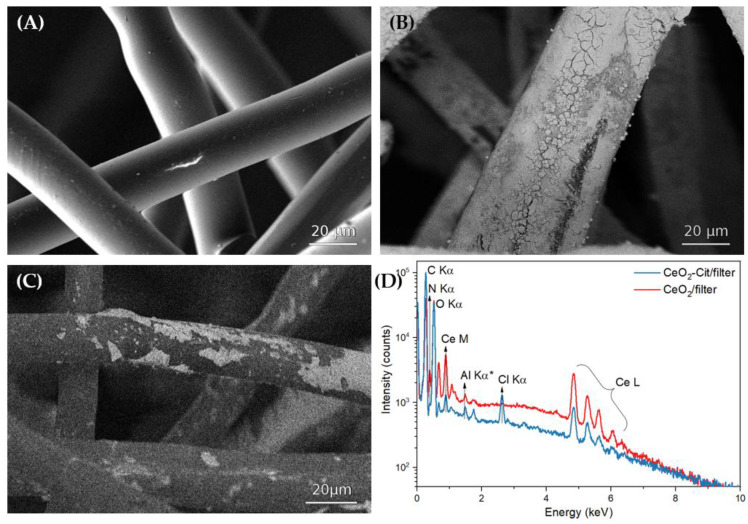
SEM images of (**A**) unmodified filter, (**B**) filter after deposition of CeO_2_ nanoparticles, (**C**) filter after deposition of CeO_2_-Cit nanoparticles. (**D**) Energy-dispersive X-ray spectra of filters covered with CeO_2_ and CeO_2_-Cit nanoparticles. * Al Kα is the impurity signal from the SEM stage.

**Table 1 nanomaterials-13-03168-t001:** Content and surface concentration of ceria after deposition onto filters.

Treatment Conditions	CeO_2_		CeO_2_-Cit	
	Content, wt.%	Surface Concentration 10^−4^ g/cm^2^	Content, wt.%	Surface Concentration 10^−4^ g/cm^2^
50 cm, 20 s	2.1	3.4	0.8	1.2
50 cm, 60 s	4.2	6.7	0.7	1.1

**Table 2 nanomaterials-13-03168-t002:** Antibacterial activity of the samples.

	CFU before Incubation	CFU after 24 h	Antibacterial Activity	Average Antibacterial Activity for This Type of Sample
Control	2.0 × 10^8^	2.2 × 10^6^	Not applicable	-
CeO_2_ 1	2.6× 10^8^	4.4 × 10^6^	No activity	0.45
CeO_2_ 2	2.5 × 10^8^	1.3 × 10^6^	0.33	
CeO_2_ 3	2.2 × 10^8^	3.7 × 10^5^	0.82	
CeO_2_ 4	2.6 × 10^8^	3.9 × 10^5^	0.87	
CeO_2_-Cit 1	2.7 × 10^8^	2.0 × 10^2^	4.17	4.15
CeO_2_-Cit 2	2.6 × 10^8^	1.5 × 10^2^	4.28	
CeO_2_-Cit 3	2.5 × 10^8^	2.5 × 10^2^	4.04	
CeO_2_-Cit 4	2.6 × 10^8^	2.3 × 10^2^	4.09	

## Data Availability

The data presented in this article are available on request from the corresponding author.

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
