# Peer review of "Coating of Filter Materials with CeO2 Nanoparticles Using a Combination of Aerodynamic Spraying and Suction"

_nanomaterials, 2023, doi:10.3390/nano13243168_

Round 1

Reviewer 1 Report

Comments and Suggestions for Authors

The paper presents research findings focused on creating a novel, cost-effective and efficient coating method for trichloroethylene filters utilised in air conditioning. The authors propose a method that incorporates aerodynamic emitters and a vacuum generation system to produce coatings with antibacterial characteristics on the filter's surface. 

The article is well-structured, presenting clear and significant relevance to the field. Scientific integrity is maintained, and the experimental design appears suitable for the study.

The reference list contains 46 publications, although 26 of these were written over five years ago, and 13 of these were written over a decade ago. The visual aids, comprising graphs, diagrams, figures and images, are presented in an easily grasped manner, without being difficult to interpret.

The paper's authors examined the impact of various technological parameters on the efficacy of ceria nanoparticle deposition, highlighting their optimal values. In subsequent stages of the study, variations in the efficacy of coating a particular commercial filter material with CeO2 and CeO2-Cit sols were observed using the above method.

These findings demonstrated that coating the filter fibre with CeO2-Cit sol resulted in almost tenfold better antimicrobial properties compared to coating it with CeO2. 

In addition to comments on the date of publication of more than half of the literature cited, the following points are somewhat unsatisfactory.

The research was carried out for only one commercially available filter material, while the conclusions are formulated for the total number of trawls used.

Due to the low degree of coverage of the filter material with CeO2-Cit sol nanoparticles, there is a lack of discussion on how to improve this efficiency to ensure the effectiveness of the use of such filters.

Insufficient coating of the filter material with CeO2-Cit sol nanoparticles is the basis of deficient discussion surrounding measures to enhance efficiency. This is necessary to guarantee the effectiveness of filter utilization.

Upon thorough examination, I am of the opinion that the current form of the manuscript is suitable for acceptance. However, I suggest that the authors review the references in their literature list to ensure the inclusion of more recent research from my perspective. 

Author Response

Dear Editor,

We would like to thank you and the reviewers for the attention you have given the manuscript. We have taken all amendments into account and tried to address them all. We are sending you our manuscript with the corrections we have made according to the comments. Below you will find a list of our comments and corrections.

Answers to Referee #1:

Issue 1. The reference list contains 46 publications, although 26 of these were written over five years ago, and 13 of these were written over a decade ago.

We thank the reviewer for the comment. We have added more recent publications to the reference list (publications number 3, 6, 14-16, 20-24, 26, 35, 36, 47 and 49).

Issue 2. The research was carried out for only one commercially available filter material, while the conclusions are formulated for the total number of trawls used.

We thank the reviewer for his attention. We have changed the conclusions and specified the type as non-woven fine filters for air conditioners (a common filter suitable for various air conditioning systems).

Issue 3. Due to the low degree of coverage of the filter material with CeO2-Cit sol nanoparticles, there is a lack of discussion on how to improve this efficiency to ensure the effectiveness of the use of such filters. Insufficient coating of the filter material with CeO2-Cit sol nanoparticles is the basis of deficient discussion surrounding measures to enhance efficiency. This is necessary to guarantee the effectiveness of filter utilization.

We appreciate the reviewer's comment. Although it is plausible that a higher CeO2-Cit concentration on the surface would enhance the antimicrobial effects of the coating, we investigated both CeO2-Cit brine with mixed oxidation state and pure CeO2 brine in our study. We decided to apply these sols to the surface of the filter fibres under identical conditions to ensure a valid comparison. Since our aim was to develop a method for the production of antibacterial filters, we focussed on the technique of ceria deposition. These results will contribute to the development of more effective antibacterial filters for practical application. Our findings demonstrate that even lower concentrations of CeO2-Cit have strong antibacterial activity, and the application method ensures an even distribution of the sol nanoparticles throughout the filter bulk and surface. An important result is also the proof that the antibacterial activity is due to the presence of trivalent cerium in the cerium oxide structure.

Note that CeO2-Cit can adhere less well to the fiber surface. Possible approaches to increase the content are surface modification, which would clearly impair the antibacterial activity of the nanoparticles, or modification of the application conditions. However, the results show that a significant increase in the deposition time only slightly increases the concentration of sol nanoparticles, making the deposition process less resource-efficient.

We have added the relevant paragraph into the results section (lines  345-355)

Please find attached the file "Response to reviewers", which dscribes all the changes done in the manuscript

Reviewer 2 Report

Comments and Suggestions for Authors

Please find it in the attachment

Author Response

Dear Editor,

We would like to thank you and the reviewers for the attention you have given the manuscript. We have taken all amendments into account and tried to address them all. We are sending you our manuscript with the corrections we have made according to the comments. Below you will find a list of our comments and corrections.

Answers to Referee #2:

Issue 1. Although the authors assumed that both improved antibacterial active and use of high-throughput and resource-saving method for deposition of cerium oxide are innovations. As the reviewer knew, the dip-rolling process from liquid has been widely used for nanoparticle deposition in various industrial sites with high throughput and environmental friendliness. The aerodynamic emitter and simultaneous suction process are involved vacuum and limited output.

We thank the reviewer for the comment. The dip-rolling process from liquid has been widely used for nanoparticle deposition in various industrial sites with high throughput. While this process for nanoparticle deposition is simple and high-speed, it has several drawbacks when used in various industrial sites. One of the main disadvantages is the lack of precise control over the thickness and uniformity of the nanoparticle coating. This can lead to variations in coating thickness across the filters, which may not meet the stringent requirements for antibacterial applications. Achieving the desired coating thickness often requires multiple dipping and rolling cycles, which can increase production time and costs, making it less suitable for high-throughput manufacturing. Furthermore, the dip-rolling process may result in inefficiencies and waste of nanoparticle materials. In summary, while the dip-rolling process has its advantages, such as simplicity and low equipment requirements, its limitations in terms of control, efficiency, and applicability to complex surfaces make it less favorable compared to other deposition methods in many industrial settings.

In addition, the need for liquid processing limits the performance of the method and requires the use of additional reagents and the additional drying of the material, thus increasing the cost of the final product. Thus, many industrial sites chose the spraying technique for their coating processes.

We have added the relevant information into the introduction. (lines 94-107)

Issue 2.  Despite the authors prosed the antibacterial performance difference of two kinds of CeO2 NPs deposited on the nonwoven fibers. The authors need to clarify this issue by measuring the antibacterial performance of pure CeO2 NPs dispersion, rather than the NPs deposited on nonwoven fabrics. At the same time, the authors may need to find out how to control the preparation process to achieve higher antibacterial efficacy of CeO2 NPs.

We thank the reviewer for the comment. It is true that in many cases the analysis of the pure substance provides more definitive information about its antibacterial properties. Cerium oxide has been intensively studied over the last decade and its properties, including those related to size and valence, are well known (See, for example, the most recent review at https://doi.org/10.1016/j.jre.2022.09.003). However, studying the antibacterial activity of a sol in solution involves a different experimental design and does not take into account the properties of the coating, so it cannot serve as a control. To analyse coating samples, we need a substrate, which was used as a filters in our work. The aim of the work was to develop a method to produce an antibacterial filter. However, the proposed manuscript describes a more practical, applied use of the antibacterial properties of CeO2.

Our work is focused on the technical details of CeO2 application, such as the penetration into the filter material and adhesion. Therefore, we believe that it is correct to compare only samples obtained under identical conditions, with an unmodified filter as a control.

We are happy that we were able to demonstrate the superiority of CeO2-Cit antibacterial action, which was expected based on the oxidation states. A truly significant achievement was the finding that coating of filters based on a cerium oxide nanoparticles with presence of trivalent cerium demonstrates significantly greater antibacterial activity, even at lower concentrations.

Issue 3. The durability of NPs on their nonwovens, or improving the adhesion of NPs in the fabrics, is one of the most important factors in the effectiveness of the filter application. The authors may need to pay attention to the improved adhesion of NPs by measuring the running time related antibacterial performance or the stability of NPs after some running time.

We thank the reviewer for this suggestion. In continuation of our previous work we have monitored the antibacterial performance of the filter materials coated with CeO2-Cit after 1 month. Over the month the filter was exposed to air, which was circled inside the laboratory air condition system. We found that there is no significant decrease in antibacterial performance (the average antibacterial activity decreased to 3.5), however, more systematic studies with a larger number of samples are needed and will be further planned.

We have added a paragraph to the results section (lines 355-359)  

Issue 4. The authors need to add more information on the details of the deposition method by explaining its working principle and characteristics, because not every reader has a good understanding of this process of producing antimicrobial nanoparticles on nonwovens.

We are very grateful for this correction. We have added the explanation (lines 150-159)

Issue 5. The authors need to make a brief literature review on the methods of deposition of antimicrobial nanoparticles on nonwovens, compare the advantages and disadvantages of different methods, and explain why the authors chose this method.

We are very grateful for this suggestion. We have added information on the dip-rolling technique, which is the main competitor of the spraying techniques (for that technique we have added reference 51), a more detailed review of the coating techniques may be found in the review (reference 44, which was added)

Please see attached the "Respose to the reviewers", where we have described all the changes

Round 2

Reviewer 2 Report

Comments and Suggestions for Authors

Reviewers are more satisfied with this revision by the authors and may consider it for publication.